# Influence of the Osteogenomic Profile in Response to Alendronate Therapy in Postmenopausal Women with Osteoporosis: A Retrospective Cohort Study

**DOI:** 10.3390/genes14020524

**Published:** 2023-02-19

**Authors:** Alejandra Villagómez Vega, Jorge Iván Gámez Nava, Francisco Ruiz González, Misael Pérez Romero, Walter Ángel Trujillo Rangel, Ismael Nuño Arana

**Affiliations:** 1Doctorado en Farmacología, Departamento de Fisiología, Centro Universitario de Ciencias de la Salud, Universidad de Guadalajara, Guadalajara 44280, Mexico; 2Centro de Investigación Multidisciplinario en Salud, Departamento de Ciencias Biomédicas, Centro Universitario de Tonalá, Universidad de Guadalajara, Guadalajara 45425, Mexico; 3Doctorado en Salud Pública, Departamento de Salud Pública, Centro Universitario de Ciencias de la Salud, Guadalajara 44280, Mexico; 4Clínica de Osteoporosis del Antiguo Hospital Civil “Fray Antonio Alcalde”, División de Medicina Interna, Guadalajara 44280, Mexico; 5Centro de Investigación Multidisciplinario en Salud, Departamento de Salud y Enfermedad, Centro Universitario de Tonalá, Universidad de Guadalajara, Guadalajara 45425, Mexico

**Keywords:** osteoporosis, alendronate, bone mineral density, single nucleotide polymorphism, osteogenic profile, personalized therapy

## Abstract

Background: Postmenopausal osteoporosis is a multifactorial disease. Genetic factors play an essential role in contributing to bone mineral density (BMD) variability, which ranges from 60 to 85%. Alendronate is used as the first line of pharmacological treatment for osteoporosis; however, some patients do not respond adequately to therapy with alendronate. Aim: The aim of this work was to investigate the influence of combinations of potential risk alleles (genetic profiles) associated with response to anti-osteoporotic treatment in postmenopausal women with primary osteoporosis. Methods: A total of 82 postmenopausal women with primary osteoporosis receiving alendronate (70 mg administered orally per week) for one year were observed. The bone mineral density (BMD; g/cm^2^) of the femoral neck and lumbar spine was measured. According to BMD change, patients were divided into two groups: responders and non-responders to alendronate therapy. Polymorphic variants in *CYP19*, *ESR1*, *IL-6*, *PTHR1*, *TGFβ*, *OPG* and *RANKL* genes were determined and profiles were generated from the combination of risk alleles. Results: A total of 56 subjects were responders to alendronate and 26 subjects were non-responders. Carriers of the G-C-G-C profile (constructed from rs700518, rs1800795, rs2073618 and rs3102735) were predisposed to response to alendronate treatment (*p* = 0.001). Conclusions: Our findings highlight the importance of the identified profiles for the pharmacogenetics of alendronate therapy in osteoporosis.

## 1. Introduction

Postmenopausal osteoporosis is characterized by a decrease in bone mineral density (BMD) and an alteration in the microarchitecture of the bone, which confers a greater risk of fractures in the event of minimal trauma [1,2]. It is known to have high heritability, and its BMD variability ranges from 60 to 85%, which may be due genetic factors [3]. More than 200 loci have been associated with the generation of more than 10% BMD variation [4]. The variations in the genes involved in bone metabolism that have been shown to generate the most significant contribution are mainly single nucleotides (SNPs) and have helped to deepen our understanding of the pathophysiology of osteoporosis [5].

These polymorphisms can contribute to causing an imbalance between bone formation and resorption [6], mostly related to the pathways of bone metabolism, such as the RANK/RANKL/OPG and Wnt/β–catenin systems [7]. In addition, other polymorphisms can influence the expression and upregulation of molecules associated with increasing bone resorption, such as IL-1, IL-6, TNF and PTH, among others; these are considered to be pro-resorptive molecules [8]. Other pathways modulate bone resorption by acting on aromatase, estrogens and TGF-β, among other molecules that can increase bone formation [9,10].

Although some contributions of these molecules are known, these may vary according to population, age and gender. The identification of SNPs and their interactions in genes involved in bone metabolism may become of great importance for the identification of people with a higher risk of suffering from osteoporosis and patients with differences in susceptibility to treatments [11].

Antiresorptive therapies are commonly used as the first pharmacological treatment for osteoporosis. These antiresorptive drugs act by modulating the activation of osteoclasts, thereby decreasing bone resorption. Although these agents have been shown to increase BMD, or at least maintain or delay the decrease in BMD, they are generally prescribed late, when significant bone mineral loss is already present [12]. Of the oral bisphosphonates, alendronate is one of the most frequently used drugs for the treatment of osteoporosis since this bisphosphonate has been demonstrated to reduce the risk of osteoporotic fractures [13]. However, 8–41% of osteoporotic patients have an inadequate response to alendronate and BMD continues to decrease after one or two years of treatment; these patients can also present with new fractures [14,15]. The lack of adherence to bisphosphonates may be a reason explaining these failures; however, some patients continue to experience osteoporotic fractures despite their excellent adherence to the treatment. This is known as an inadequate response (IR) to bisphosphonates [16,17,18].

Pharmacogenetic studies are relevant since the genetic determinants of the failure of bisphosphonates are not clear [19]. These studies could help in predicting the possible response to antiresorptive treatment. There is information suggesting that genetic factors account for approximately 15 to 30% of personal responses to drugs in the metabolism, and up to 95% of the variability in disposition [5]. Due to the high rate of therapeutic failure to antiresorptive therapy with alendronate, the lack of knowledge regarding the causative factors of this and the clinical impact of SNPs in the response to these drugs, it is necessary to generate new knowledge in this field. Therefore, we investigated the association between some risk alleles (genetic profiles) associated with BMD and the response to alendronate treatment in women with postmenopausal primary osteoporosis.

## 2. Materials and Methods

### 2.1. Subjects

This study included 82 postmenopausal Mexican mestizo women with primary osteoporosis. Patient recruitment was from January 2017 to July 2019. These patients were referred by an osteoporosis clinic at a civic hospital (Antiguo Hospital Fray Antonio Alcalde) in Guadalajara, Mexico. The inclusion criteria for participants were as follows: (1) had a diagnosis of osteoporosis according to the clinical practice guidelines for the diagnosis and management of osteoporosis [20]; (2) aged 50 years or older; (3) included in the cohort at the time when they received alendronate (70 mg administered orally once per week) for one year (according to the clinical guidelines for pharmacological treatment of the National Osteoporosis Foundation, USA); (4) received calcium and vitamin D supplementation therapy; (5) had a clinical chart available at the hospital. The exclusion criteria were patients receiving parathyroid hormone therapy, selective estrogen receptor modulators (SERMs), glucocorticoids or biologic therapy, or presenting a secondary comorbidity associated with low BMD (such as thyroid disease or chronic renal failure) or the presence of alcoholism. Only one patient with osteoporosis per family was allowed.

Study development: All the subjects underwent a physical examination and completed a detailed questionnaire on family and medical histories, as well as on lifestyle habits. Body mass index (BMI) was calculated as weight (kg) divided by height squared (m^2^). Treatment adherence was determined using the questionnaire validation of the Adherence Evaluation of Osteoporosis (ADEOS) treatment for osteoporotic postmenopausal women [21]. All patients included in the study showed adequate adherence to the treatment.

### 2.2. Bone Densitometry Measurements

The BMD (g/cm^2^) of the lumbar spine (L1–L4) and total femoral neck was determined using dual-energy X-ray absorptiometry (DXA) with a Lunar Prodigy Advance densitometer (GE Medical Systems Lunar ver. 16. software; GE Medical Systems, Madison, WI, USA).

The BMD of the lumbar spine and total femoral neck were classified according to the guidelines of the International Society for Clinical Densitometry (ISCD, 2013) [22]. These guidelines recommend that postmenopausal women be classified as having osteoporosis using T-scores, i.e., when the DXA results of the lumbar spine or total femoral neck show a decrease of less than 2.5 standard deviations (SD). The variation coefficient of the BMD measurements in our institution was 1.2% at the lumbar spine and femoral neck. During the follow-up period, each participant had repeated BMD measurements approximately once a year, performed by the same technician, using the same machine and under the same standard operating procedure.

### 2.3. Isolation of Genomic DNA 

Genomic DNA was extracted from 82 unrelated volunteers; DNA was extracted from a more significant part of the FTA Whatman^®^ card using automated protocols [23,24]. The subjects provided their signed informed consent. The sample collection was approved by the Ethics Committee for Research of the Hospital Civil of Guadalajara, Jalisco, Mexico.

### 2.4. SNP Genotyping 

The genomic DNA was diluted to 20 ng/µL and collected in propylene cryotubes with a capacity of 200 µL (Eppendorf™) as the working samples. The genotyping of polymorphisms was performed using quantitative polymerase chain reaction (qPCR). Several SNPs were analyzed with specific primers and TaqMan probes (TaqMan SNP genotyping assays; Applied Biosystems, Foster City, CA, USA). The following TaqMan ID protocols were followed: C_8794675_30, C_3163591_10, C_1839697_20, C_9698268_1, C_1971046_10, C_1971047_40, C_8708473_10 and C_30009803_10 (Applied Biosystems); the StepOne™ Real-Time polymerase chain reaction (PCR) system was employed for this purpose (Applied Biosystems) [25]. Information about SNPs from the *CYP19*, *IL-6*, *ESR1*, *PTHR1*, *TGFβ*, *OPG* and *RANKL* genes was obtained from the SNP HapMap database https://www.ncbi.nlm.nih.gov/snp (accessed on 1 July 2016). The SNPs for the study were selected according to the following criteria: the important roles they play in bone metabolism and previously demonstrated associations with decreased BMD and fracture [4,26], and minor allele frequency higher than 10% in study populations. Eight markers were selected: rs700518, rs1800795, rs9340799, rs724449, rs1800469, rs3102735, rs2073618 and rs9533156 [27,28,29]. We aimed to identify profiles in order to help improve the management of the disease by generating personalized treatments.

### 2.5. Response to Treatment 

Non-response to treatment was defined according to the UK’s National Institute for Health and Care Excellence guidelines: the occurrence of a new fragility fracture despite full adherence to treatment for one year or a decrease in BMD to a value below the baseline value [30]. Total femoral neck BMD (g/cm^2^) was measured at baseline and at least after 12 months of treatment. An increase of 2% in BMD to a value above the baseline level was considered an acceptable response to alendronate therapy, while a decrease in BMD was considered as treatment inefficiency. According to this, the patients were classified as responders or non-responders to alendronate therapy [16].

### 2.6. Statistical Analysis

Data analysis was performed using SPSS Statistics 21.0 for Windows (SPSS, Chicago, IL, USA) and SNP Analyzer 2.0. We used the Hardy–Weinberg equation to calculate the expected genotype frequencies for each polymorphism, which were then compared with the observed values using the chi-squared test. The expected results are shown with 95% confidence intervals (CI). The Kolmogorov–Smirnov test was used to assess the normality of the data distribution; the data were found to be normally distributed. Descriptive statistics are presented as the mean ± SD. We used paired *t*-tests to compare the BMD in g/cm^2^ (total femoral neck) obtained at baseline and after one year of treatment with alendronate. The response to alendronate therapy was evaluated according to the change in total femoral neck BMD. A 2% increase in BMD, defined as a significant change according to the concept of the trend assessment margin (TAM), was considered as indicative of a response to the alendronate treatment, while a decrease in BMD was considered as indicative of treatment inefficiency. Allelic combinations (profile) were constructed from the allelic frequency and analyzed for their association with response to treatment using SNP Analyzer. The differences in BMD at baseline and changes after one year of treatment (response to treatment) among the profile groups were analyzed using the Pearson chi-squared test and compared using the generalized linear model (GLM). The participants were classified into two groups: responders and non-responders. We used odds ratios to identify the association between genotype and allele profiles and the response to treatment. *p*-values < 0.05 were defined as statistically significant. We used Bonferroni correction to adjust the biases inherent with multiple testing.

## 3. Results

### 3.1. Basic Characteristics of Study Subjects

We evaluated 82 postmenopausal female patients with primary osteoporosis. All patients completed the treatment of 12 months of 70 mg of alendronate weekly and had BMD measurements at baseline and after 12 months of treatment.

Table 1 presents the epidemiological characteristics of the included women. These patients had a mean age of 65 ± 8.9 years, and smoking was reported in 20% of patients. The mean BMI was 26.1 ± 3.9, with 44% having an average or low weight and 56% being overweight or obese. Based on the treatment outcomes, 56 subjects were classified as responders to alendronate and 26 subjects were classified as non-responders to alendronate therapy. The inadequate response frequency was 31.7%.

### 3.2. Allele and Genotype Frequencies and Profile Structure

The genotyping of all 82 participants was successful. Table 2 shows the allele and genotypic frequencies and the Hardy–Weinberg equilibrium (HWE) analysis of the SNPs analyzed. Seven profile blocks from eight SNPs generated from the allele combinations were analyzed, with the total frequency being higher than 5%. The presence of these seven profiles was found in 87.8% of all patients.

### 3.3. SNPs and Profiles Associated with the Alendronate Treatment Response

The genotype association with response to treatment is shown in Table 3. No single SNP was found to be associated with response to alendronate treatment.

Table 3 shows the associations of genotypes with response to treatment. We did not find any associations with response to alendronate treatment. 

Seven profiles with total frequencies higher than 5% are shown in Table 4. The allelic combinations were constructed from the genotyping of eight SNPs: rs700518, rs9340799, rs1800795, rs724449, rs3102735, rs2073618, rs1800469 and rs9533156. 

The table shows the different profile frequencies divided into two groups according to the response to treatment and the comparison of these frequencies. Additionally, profile association analyses with the response to treatment frequency and the risk analysis for each profile are shown. The profiles did not show any association with the treatment response, except for profile 2 (*p* = 0.001); we did not consider the OR due to the small sample size.

Figure 1 shows the comparison of frequencies based on the response to treatment in the presence or absence of profile 2, where it can be seen that 100% of the patients who carried profile 2 (*p* = 0.001) responded to treatment, while 31% of the patients who did not carry profile 2 did not respond to treatment. 

## 4. Discussion

The objective of pharmacological treatment in osteoporosis is to reduce the fracture risk, which is related to an increase in BMD, or at least to a delay in BMD loss [17]. Osteoporosis treatments can be divided in drugs that increase bone formation by stimulating osteoblast activity and drugs that decrease bone resorption by inhibiting osteoclast activity [31].

Alendronate is the main bisphosphonate used as the first-line treatment for patients with osteoporosis, as it has been shown to increase BMD and reduce the fracture risk [26]. However, the causes responsible for non-response to this drug are unknown. A meta-analysis conducted by Sebba [15] found that the frequencies of an inadequate response to alendronate, risedronate and ibandronate in osteoporosis ranged from 8% to 25%. Francis [32] showed a frequency of inadequate response to bisphosphonates of 15% in osteoporosis patients. However, this inadequate response may be more frequent in clinical practice. Watts et al. [33] demonstrated an inadequate response to bisphosphonates ranging between 10% and 50%. However, these authors were unable to identify any risk factor for inadequate response upon clinical examination.

Genetic factors account for up to 85% of BMD variability and they are not currently considered when deciding on a pharmacological treatment. There is evidence that SNPs are relevant genetic factors involved in the pathophysiology of osteoporosis [34]. Morris et al. [4] performed a meta-analysis of genome-wide association studies (GWAS) and identified variations in more than 200 genes involved in bone metabolism and associated with lower BMD and fractures.

Marozik et al. [35], in a cohort study, found that osteoporosis patients who did not respond to bisphosphonate treatment (40%) had a higher proportion of gene variants compared to patients showing response to this drug. This result is similar to the present study, where we found an inadequate response to alendronate in 32% of patients.

The generation of profiles from gene variants can contribute to detecting genes involved in the pathophysiology of a disease and to detecting different responses to pharmacological therapies [5]. Hopwood et al. [6] generated different expression profiles associated with the activation of genes (150) during the bone repair process in patients with osteoporosis who had a femur fracture; the main genes involved were related to the activation and maturation of osteoclasts. Marozik et al. [35] performed a cohort study in which they looked for an association between response to treatment and combinations of allelic variants of markers associated with the disease, and found that carriers of one combination were predisposed to a negative response to bisphosphonate therapy and carriers of a different combination were overexpressed in responders. However, in our study, the analysis of the alleles considered to be risk factors for inadequate response to alendronate treatment did not show these associations. By generating profiles from a combination of risk alleles, we found a significant difference in patients who had profile 2, who all responded to treatment with alendronate (Table 4). In the present study, none of the alleles were considered to be risk factors for osteoporosis or showed an independent association with the therapeutic response to alendronate, supporting the notion of an additive effect in the interaction of these alleles.

In the present study, profile 2 was formed by the SNPs rs700518 of the *CYP19* gene, rs1800795 of the *IL-6* gene, and rs2073618 and rs3102735 of the *OPG* gene. These reported frequencies coincide with those published in the literature for the SNPs rs1800795 [28,29] and rs2073618 [36] in the Mexican population, while rs3102735 [37] was consistent with a Caucasian population. In addition, they were found using the HWE (Table 2).

The CYP19 variant generates a change in an amino acid that decreases the activity of the enzyme [38]. The IL-6 variant is associated with an increase in its levels; it is known that this cytokine acts as a pro-resorptive factor by favoring the release of RANKL and increasing osteoclastogenic activity [39,40]. In addition to this, OPG variants are associated with a lower activity of the protein, which decreases its activity by acting as a decoy and preventing the binding of RANKL to its receptor in osteoclasts. This could generate more significant maturation and activation of osteoclasts, meaning that bone resorption would also increase [41,42]. These effects alter the balance between bone formation and resorption, decreasing the BMD, meaning that people with this profile have a greater susceptibility to osteoporosis and a higher risk of fracture. Patients with this profile could likely have an imbalance in bone metabolism and presumably more significant osteoclastogenic activity, so treatment by antiresorptive drugs such as alendronate would act by inhibiting osteoclasts, allowing for an increase in BMD. However, we also know that there are other factors that can influence bone metabolism.

As was observed in this study, it is important to clarify that the effect of a putative risk allele of an SNP is very small. As we know, multiple factors are involved during the control and processing expression of a gene, and these are influenced by many other factors, such as epigenetic mechanisms. It would be very difficult to directly explain its effect on the phenotype, or in this case, on the response to management. Therefore, more detailed studies are required to analyze the gene–gene effect, so far little studied and the mechanisms that interact to support these suggestive results.

Although the frequencies of these risk alleles are representative of studies reported in the literature for the Mexican population, by stratifying the sample by profile, the number of patients per group decreased considerably, which made it difficult to establish an OR (Table 4).

There are models that help predict responses to treatment, but they only take lifestyle, BMD and anthropometric characteristics into account. By not taking genetic factors into account, the predictive ability is low. A better evaluation should take gene–gene and gene–environment interactions into account [34].

The present study evidences that the environmental factors did not significantly modify the response to treatment, so these data support the hypothesis that genetic factors could be the primary determinant (Table 1).

All patients received calcium and vitamin D supplementation during alendronate treatment. Low vitamin D levels have been associated with an inadequate response to alendronate [43]. Therefore, in would be important to reduce the environmental influence factor of diet, which could be a bias factor and modify the response to treatment. Treatment adherence is essential to obtain results, such as an increase in BMD, and to reduce the risk of fracture. Our study clearly shows the importance of considering genetic factors when deciding on the most appropriate treatment for patients while aiming to reduce the frequency of IR.

Whether or not response to treatment occurred is determined after receiving treatment for one year, so if it does not work, this represents lost time for the patient, and the finances and the quality of life of the patient may be affected by not receiving adequate treatment. The results of this study may help to guide physicians to personalize treatments more suitably.

The main limitation of our study is the small number of samples analyzed and the minimum follow-up period to observe the response to treatment. A larger sample size could help to establish associations with different profiles. Another limitation is the lack of OR, which could be used to determine the possible effect that the profiles analysis may have on these patients. Another one is the low number of SNPs analyzed, so it is essential to include more genes that act in different bone metabolic pathways in future. In addition, a depth analysis that includes the genotypes, instead of the allelic markers, could be more informative in providing knowledge regarding the entire genetic effect. In addition, the pharmacological management of the disease should be included.

An interesting fact is shown in Table 4, where a high frequency of profile 1 is shown (in 68% of patients), which is the same as profile 2 (in 12% of patients), except for the addition of the ESR1 risk allele. This did not show an association with the response to treatment. This can be explained by the critical contribution of ESR1 in the development of the disease, as it has been found to be one of the most highly involved markers [44]. Therefore, its high frequency, as reflected in the Hardy–Weinberg disequilibrium, was expected (Table 2). We suggest that further studies be carried out in order to determine the effects of genetic factors, gene–gene interactions, and further alternative implementations as strategies for personalized antiresorptive treatment.

## 5. Conclusions

The presence of profile 2 suggests a possible influence on being predisposed to positive response to alendronate therapy in Mexican women. The analysis of osteogenomic profiles can help to find altered metabolic pathways and, thus, enable individualized therapy in order to increase treatment efficacy and reduce the percentage of IR in patients with osteoporosis.

## Figures and Tables

**Figure 1 genes-14-00524-f001:**
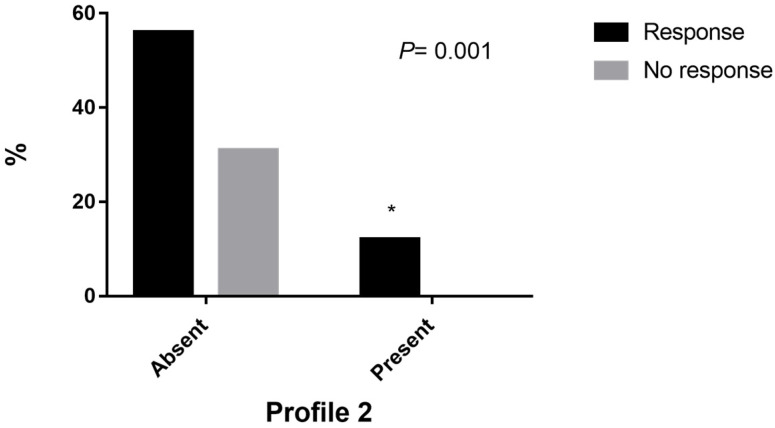
Frequencies based on the response to treatment of profile 2. * (*p* = 0.001).

**Table 1 genes-14-00524-t001:** Characteristic of osteoporosis patients.

Variable	OP Osteoporosis n = 82	
Sex Female, n (%)	82 (100)	
Smoking, n (%)		
Never	57 (69.6)	
Former smoker	8 (9.7)	
Current	17 (20.7)	
Age (years), mean ± SD	65 ± 8.9	
Body Mass Index (kg/m^2^), mean ± SD	26.1 ± 3.9	
Normal or low weight, n (%)	36 (43.9)	
Overweight and obesity, n (%)	46 (56.1)	
Menopause, n (%)	82 (100)	
BMD		
	Base line	12 months
BMD Hip	0.818 g/cm^2^ ± 0.090	0.850 g/cm^2^ ± 0.133
BMD L1–L4	0.768 g/cm^2^ ± 0.075	0.800 g/cm^2^ ± 0.080
Treatment		
Monotherapy with alendronate	82 (100)	
Results of the BMD at year		
Nonresponse to alendronate	26 (31.7)	
Response to alendronate	56 (68.3)	

Abbreviations: BMD, bone mineral density.

**Table 2 genes-14-00524-t002:** Genotype and allele frequencies of polymorphisms.

Polymorphism	Allele Frequency (%)	Genotype	Number	Genotype Frequency	HWE *p* Value
rs700518	A = 18.3 C G = 81.7 T	AA AG GG	4 22 56	4.9 26.8 68.2	X^2^ = 0.384 *p* = 0.535
rs9340799	A = 58.7 G G = 41.3 A	AA AG GG	21 46 8	28 61.3 10.7	X^2^ = 5.253 *p* = 0.021
rs1800795	G = 89.6 C = 10.4	GG GC CC	66 15 1	80.5 18.3 1.2	X^2^ = 0.172 *p* = 0.678
rs724449	C = 43.3 A T = 56.7 G	CC CT TT	11 49 22	13.4 59.8 26.8	X^2^ = 3.862 *p* = 0.0490
rs3102735	A = 77.2 T G = 22.8 C	AA AG GG	49 27 5	60.5 33.3 6.2	X^2^ = 0.058 *p* = 0.809
rs2073618	G = 74.4 C = 25.6	GG GC CC	43 33 4	53.7 41.3 5	X^2^ = 0.540 *p* = 0.462
rs1800469	G = 61.5 T A = 38.5 C	GG GA AA	30 36 12	38.5 46.1 15.4	X^2^ = 0.049 *p* = 0.825
rs9533156	T = 92 C = 62	TT TC CC	22 48 7	28.6 62.3 9.1	X^2^ = 6.743 *p* = 0.0094

Abbreviations: HWE, Hardy–Weinberg equilibrium (the chi-square test value).

**Table 3 genes-14-00524-t003:** SNP association between response to alendronate treatment.

SNP	Genotype	Response		
		Si	No	*p*-Value
rs700518	AA	1	3	0.116
	AG	14	8	
	GG	41	15	
rs9340799	AA	14	7	0.477
	AG	32	17	
	GG	10	2	
rs1800795	GG	1	0	0.697
	GC	11	4	
	CC	44	22	
rs724449	CC	6	5	0.223
	CT	37	12	
	TT	13	9	
rs3102735	AA	3	2	0.749
	AG	20	7	
	GG	33	16	
rs2073618	GG	2	2	0.365
	GC	21	13	
	CC	32	11	
rs1800469	GG	17	13	0.173
	GA	27	10	
	AA	10	2	
rs9533156	TT	4	4	0.538
	TC	34	15	
	CC	16	7	

Abbreviations: SNP, single nucleotide polymorphism.

**Table 4 genes-14-00524-t004:** Frequency and analysis profile.

Profile	Response (%)	No Response	*p*-Value	OR	*p*-Value
G_1_G_2_C_3_A_5_C_6_ (1)	30 (68.2)	14 (31.8)	0.981	1.005	0.98
G_1_C_3_G_5_C_6_ (2)	10 (100)	0	**0.001 ***	-	0.021
G_1_C_3_A_6_C_8_ (3)	10 (66.6)	5 (33.4)	0.881	1.007	0.881
G_1_C_3_A_5_G_6_ (4)	6 (66.6)	3 (33.4)	0.912	1.007	0.912
G_1_C_3_C_6_ (5)	9 (60)	6 (40)	0.445	1.436	0.445
C_3_A_5_G_7_ (6)	5 (50)	5 (50)	0.185	2.154	0.185
G_1_C_3_G_5_ (7)	13 (86.7)	2 (13.3)	0.078	0.331	0.091

Notes: 1: rs700518, 2: rs9340799, 3: rs1800795, 4: rs724449, 5: rs3102735, 6: rs2073628, 7: rs1800469, 8: rs9533156; significant values (*p* < 0.05) are presented in bold. * compared with GLMs linear model; OR, odds ratio.

## Data Availability

Not applicable.

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
