# Peer review of "Influence of the Osteogenomic Profile in Response to Alendronate Therapy in Postmenopausal Women with Osteoporosis: A Retrospective Cohort Study"

_genes, 2023, doi:10.3390/genes14020524_

Round 1

Reviewer 1 Report

In this study, the authors investigated the effect of combinations of genetic profiles associated with the response to antiosteoporotic treatment in a cohort consisting of 82 postmenopausal women with primary osteoporosis receiving alendronate for one year. The manuscript is well-written and reports interesting data.

The manuscript contains few grammatical errors that need to be corrected such as:

In Page 1, Line 21: Alendronate are used please correct it to “Alendronate is used”.

Author Response

Point 1: Alendronate are used please correct it to “Alendronate is used”.

Response 1: the sentence was edited

Reviewer 2 Report

SPECIFIC COMMENTS:

  1. In order for this study to be validated and the results to be reproducible, detailed explanation is needed in the methods on the SNP selection process. While lines 143-147 provide a generic overview, there are no details or references. Were only the 8 SNPs listed in Table 2 analyzed? If so, this should be specified in the methods along with the reason for their inclusion. Please add references as appropriate. 
  2. Abstract needs to be revised, and choose one style of presentation - structured or unstructured.
  3. In the abstract, it is mentioned that individuals with profile A-C-T-C were responders (profile 2); however, in table 4, Profile 2 is indicated as GCGC. Please check and edit as needed.
  4. Table 4: (a) How were profiles generated? Please explain this better in the results. (b) In the table, please also include the SNPs from which each profile is generated, otherwise, it is hard to make sense of individual profiles. (c) Since total number of responders and non-responders exceeds the total sample size, does that imply an individual could have more than one genetic profile? This has to be clarified, and the potential interactions discussed.
  5. The IL6 (rs1800795) polymorphism is associated with the risk of developing, prognosis of, or the treatment response to a multitude of disorders: potential associations with heart disease, Kaposi's sarcoma, type-2 diabetes, stroke, obesity, Hodgkin's lymphoma, sudden infant death syndrome, cancer (including breast cancer, gastric cancer, prostate cancer), hypertension, periodontitis, and complications arising after organ transplantations or grafts (https://www.snpedia.com/index.php/Rs1800795). Similarly, while CYP19A1 polymorphisms are noted in many osteoporosis studies, it is also studied as a potential modifier in many other estrogen-related conditions. The interactions among genes and their concerted effect is more complex, so it is imperative to remember that a conclusion just based on the statistically significant difference in the frequency of the genotypes is hasty. The discussion and conclusion sections should be revised with this in mind.
  6. Lines 323-325: As a follow up, this conclusion inflates the results observed in the study. While there is suggestive influence of genetic factors in the response to osteoporosis treatment, there is no conclusive evidence that they are in fact the primary determinant.
  7. Lines 333-334: Do the authors advocate for genotyping patients to determine if they carry profile 2 to determine whether they should be treated with alendronate? If yes, more convincing evidence must be generated. If not, this sentence must be edited.
  8. Lines 335-340 - Clarity on the limitations is needed. Explain why results from this study cannot be conclusively utilized for precision medicine. Also discuss the impact of vitamin supplementation in the BMD differences.
  9. Lines 348-349: Include that number of individuals with profile 2 and the total number of subjects, and mention the lack of OR due to small sample size.  

OTHER MINOR COMMENTS:

  1. Revise lines 41-43: "It is known to have high heritability, so genetic factors play an essential role in contributing to BMD variability, which ranges from 60 to 85%" -- this sentence is confusing as the range refers to the influence of genetic factors; however, the current sentence conveys that BMD ranges from 60 - 85%.
  2. Revise lines 43-44: "More than 200 loci have been associated with BMD, which generate variation more than 10%" -- sentence not clear.
  3. Revise lines 75-76. Also, the cited reference does not sufficiently explain the point being made.
  4. Lines 92-93: Add reference to the Clinical practice guidelines
  5. Please clarify if calcium and vitamin D supplementation was required as part of the inclusion criteria. If they were (and not part of the exclusion), comment on how this may have impacted your results.
  6. Revise lines 109-110: sentence not clear
  7. Revise lines 150-152: sentence not clear
  8. Lines 219-220 and 221-222 are redundant.
  9. Table 1: It will be useful for the reader if both mean BMD at base line and at one year are reported.
  10. Table 5 chi-squared test and OR results may be combined with table 4; however, the percentages are different between the two tables. What do the percentages indicate? It may useful to include the denominator.
  11. Line 272: "Marozik et al. (2019), in a cohort study…" is not cited
  12. Please use the HGNC gene symbol for genes or provide them in parenthesis - CYP19A1 and TNFRSF11B (https://www.genenames.org/)
  13. Line 345: Cite reference for ESR1 marker.

Round 2

Reviewer 2 Report

I appreciate the author's efforts in implementing the edits. I have a couple of comments that still need addressing by authors.
